# REVISITING LARGE LANGUAGE MODEL PRUNING USING NEURON SEMANTIC ATTRIBUTION

## ABSTRACT

Pruning large language models (LLMs) is an effective way to reduce computation while maintaining strong performance. While some studies have explored how pruning affects different tasks, most remain narrow in scope and overlook interpretability-driven analysis. In this work, we propose Neuron Semantic Attribution (NSA), a novel method that analyzes pruning through neuron-level semantics. NSA links neurons to task-relevant concepts, providing a fine-grained understanding of how pruning impacts model behavior and causes task-specific sensitivity. We also conduct a comprehensive empirical study across 24 tasks spanning diverse domains, examining the effects of pruning configurations, calibration data, and sparsity levels. Our findings demonstrate that NSA serves as a reliable tool for interpreting and guiding pruning, helping to bridge the gap between model compression and interpretability.

## 1 INTRODUCTION

Large Language Models (LLMs), such as GPT-3 and 4 Achiam et al. (2023), BERT Devlin (2018), and LLaMA families Touvron et al. (2023) have made significant strides in a wide range of Natural Language Processing (NLP) tasks, including text generation, sentiment classification, machine translation, and question answering. The advancements made by LLMs are not only transformative for the AI community but also promising in improving productivity, enabling new technologies, and improving decision making in many sectors. However, despite their extraordinary capabilities, LLMs come with a clear drawback—high computational cost during inference. Deploying these models for inference is resource-intensive, requiring substantial memory and processing power.

To address this challenge, model pruning is an effective way of accelerating LLMs. It involves removing unnecessary weights or neurons from the model, thereby reducing its size and computational cost. Based on the granularity of the components being removed, pruning methods can be categorized as structured, unstructured, and semi-structured pruning. Structured pruning is a method that remove groups of weights together. Unstructured pruning operates at the individual weight level. Semi-structured pruning focuses on enforcing N:M sparsity patterns, where within each group of M weights, only N are retained—this structured sparsity enables efficient hardware acceleration by reducing computational complexity while preserving model performance. Some recent pruning methods Frantar & Alistarh (2023); Sun et al. (2023); Zhang et al. (2024) for LLMs work without retraining the model after pruning. These methods are practical and effective ways to reduce the size of large language models while keeping good performance.

Recent pruning methods Frantar & Alistarh (2023); Sun et al. (2023); Zhang et al. (2024) have shown promising results in compressing large language models (LLMs), but they suffer from several limitations. First, these methods are often evaluated on a limited range of tasks and datasets, making it unclear how well they generalize across diverse scenarios. Second, the effects of different pruning configurations—such as sparsity levels and pruning types—are rarely explored in a systematic way. Third, existing studies largely overlook interpretability, offering little insight into how pruning alters model behavior. For example, prior works like Bandari et al. (2024) evaluated only two pruned models on nine datasets across three tasks, while Williams & Aletras (2024) examined nine models on ten datasets across four tasks. However, these analyses are still constrained in both task diversity and model coverage, particularly lacking in tasks that are sensitive to semantic richness and emotional nuance.

To address these limitations, we conduct a large-scale empirical study and introduce a neuron-level interpretability framework for analyzing pruning effects. Our evaluations span 10 LLMs, 24 datasets, 4 pruning configurations, and 4 major task categories, offering a broader and deeper understanding of pruning generalizability. Additionally, we present an interpretability method designed for pruned models, which associates neurons with task-relevant semantics and explains pruning-induced behavior changes.

We also extensively evaluate the components of pruning methods including calibration data, sequence length of tokens and target sparsity ratio of pruning methods. Although some studies Bandari et al. (2024) have begun to uncover the significant impact of the calibration set, these explorations are often limited in scope. As such, a systematic evaluation that thoroughly investigates the effects of various pruning configurations—including but not limited to calibration data—on model generalization remains a critical gap in the literature. Moreover, our results reveal that the performance on certain tasks, particularly sentiment classification, varies substantially with changes in the pruning method Frantar & Alistarh (2023); Sun et al. (2023); Zhang et al. (2024), rather than a simple uniform drop. For example, our extensive evaluations find the accuracy of the LLaMA-3-8B model pruned with Wanda drops from 77.42% to 55.08% on the Yelp dataset for sentiment classification. This observation motivates a deeper investigation into the mechanisms underlying such variability.

Motivated by some research of general model interpretation Bricken et al. (2023); Bills et al. (2023), we propose **Neuron Semantic Attribution (NSA)**, a method to assign interpretable semantics to neurons, enabling the analysis of how pruning affects the internal behavior of large language models. While prior work Dai et al. (2021); Cunningham et al. (2023) has explored neuron-level interpretability in general settings, our work is the first to apply such analysis specifically to pruned models. NSA connects neuron activation changes after pruning with task-relevant semantics, offering a direct explanation for performance shifts caused by pruning.

Through this process, NSA not only quantifies the semantic loss caused by pruning but also provides visual and token-level attribution that explains model behavior across tasks. Our analysis reveals that NSA is particularly valuable for understanding performance variance in semantically sensitive tasks such as sentiment classification. Overall, NSA enables a deeper understanding of pruning effects and provides actionable insights for designing more reliable pruning strategies.

We further propose an activation-guided pruning method that leverages low-activation neurons for masked regularization. This approach promotes sparsity in less active regions while preserving important computation pathways, enabling more targeted and interpretable pruning.

This study makes several significant contributions to the field of model optimization for LLMs:

1. We conduct the most comprehensive evaluation to date of pruning methods across 10 models, 24 datasets, 4 major task categories and 4 sparsity types. Our analysis covers the effects of calibration data, input sequence length, and sparsity ratio on pruning performance, revealing strong task-dependent sensitivity—particularly for sentiment classification and semantic similarity tasks. These findings offer new insights into the generalizability and robustness of pruned models and provide actionable guidance for future pruning design and evaluation.

2. We introduce **Neuron Semantic Attribution (NSA)**, the first interpretability framework specifically designed for pruned language models. NSA links activation changes at the neuron level with task-relevant semantics, providing fine-grained, interpretable visualizations of how pruning alters a model's internal behavior. This offers a reliable interpretability method that can help assess and guide the design of future pruning algorithms by revealing how pruning affects task-relevant semantics.

3. We provide systematic experiments that verify the crucial role of calibration data in determining the generalizability of pruned models. Our results show that pruning outcomes are highly sensitive to the distribution of calibration samples, highlighting the importance of carefully selecting and designing calibration data when applying pruning in practice. In addition, we introduce activation-guided pruning to further investigate the impact of neuron activation on pruning effectiveness.

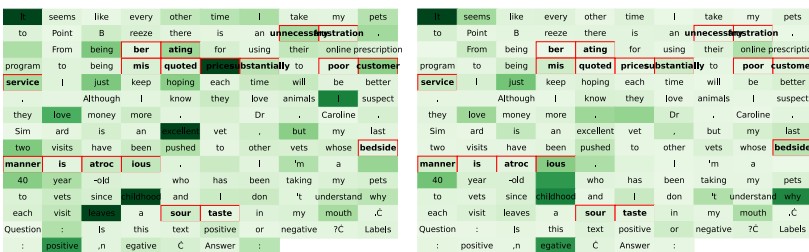

Figure 1: Visualization using NSA method on Yelp data, showing the activations of neuron 145 in layer 15 of our network. Darker colors represent stronger activations. For example, the sumed activation of the pruned and unpruned models for the sentiment-related semantics drops from 5.43 to 3.15, explaining the degraded perforamnce of sentiment classification.

## 2 RELATED WORK

**Pruning Strategy** Model pruning has emerged as an effective and efficient method for compressing large neural networks, particularly in the context of LLMs. Recent advancements in pruning methods for LLMs have introduced several promising methods. For instance, SparseGPT Frantar & Alistarh (2023) introduces a pruning method that incorporates OBS updates (optimal block sparsity), which allows for efficient pruning while retaining high accuracy in large models. This method revolutionized pruning by addressing both sparsity and computational efficiency in one framework. Following this, Wanda Sun et al. (2023) introduces a simpler, yet highly effective pruning technique that further reduces the computational burden by refining the selection of parameters to prune, while still maintaining the model's original functionality. RIA Zhang et al. (2024) improves the pruning metric with relative importance and used channel permutation to adjust the pruned model. However, it is not clear the generaliaility of those methods for downstream tasks.

**Calibration Data** Calibration data refers to a small set of unlabeled samples used to adjust model parameters during model compression including model pruning and post-training quantization. In pruning process, calibration data helps guide the selection of parameters to prune, ensuring that critical model components are preserved. Studies Frantar & Alistarh (2023); Sun et al. (2023); Zhang et al. (2024) combine calibration data with model weight to calculate the pruning metrics. In quantization Frantar & Alistarh (2022), calibration data is utilized to set activation ranges, ensuring that the quantized model maintains performance close to the original. One work Hubara et al. (2021) demonstrated that even with a minimal calibration set, it is possible to achieve accurate post-training quantization by optimizing layer parameters over this data.

**Generalizability** It is interesting to explore the generalizability of pruned models. Recent studies Williams & Aletras (2024); Bandari et al. (2024) have shown that calibration data can significantly impact the pruning process and pruned models' generalizability over various tasks. These findings suggest that the pruning metrics derived from different calibration data may lead to varied outcomes, especially when applied to diverse downstream tasks like sentiment analysis and question answering. Despite the existing study Williams & Aletras (2024); Bandari et al. (2024) on generalizability, more extensive evaluations and analysis are needed. Our work conduct extensive experiments with a variety of models, datasets, and tasks to explore the generalization capability of pruned models. Furthermore, we build upon these insights by introducing a novel interpretability method that helps to better understand these impacts, which named Neuron Semantic Attribution. This method allows for in-depth analysis of neuron-level activations and their relationships with input tokens, offering new insights into the specific components of pruned models that contribute to model predictions.

## 3 METHODOLOGY

In this study, we employed three state-of-the-art post-training pruning methods: SparseGPT Frantar & Alistarh (2023), Wanda Sun et al. (2023), and RIA Zhang et al. (2024). These methods are designed to reduce the size of large language models (LLMs) while maintaining their performance

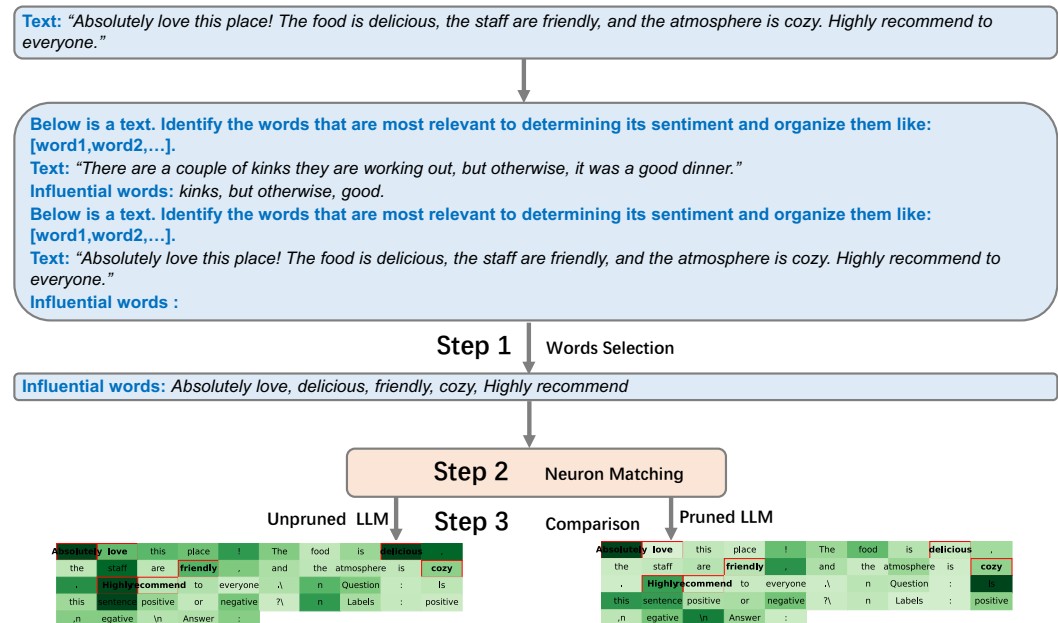

Figure 2: The framework of NSA. Step 1, Words Selection; Step 2, Neuron Matching; Step 3, Comparison.

across various tasks. Even though all three methods prune the model using a combination of weights $W$ and activations $X$, they each use different metrics $M$ and strategies to decide which parameters to prune.

### 3.1 PRUNING METHODS

- **SparseGPT**: This method selects parameters based on the metric: $M = \frac{W^2}{[H^{-1}]^2}$, where $H^{-1} = (XX^\top + \lambda I)^{-1}$. After pruning, SparseGPT also applies an OBS Hassibi et al. (1993) update to compensate for the pruned parameters.

- **Wanda**: This method uses a simple yet effective metric for pruning: $M = |W| \cdot \|X\|_2$. After pruning, there is no need for any parameter update step.

- **RIA**: This method calculates the pruning metric using relative importance: $M = \left( \frac{|W_{ij}|}{\sum |W_{*j}|} + \frac{|W_{ij}|}{\sum |W_{i*}|} \right) \times (\|X_i\|_2)^a$. After calculating the metric, channel permutation is used to adjust the model for N:M sparsity.

These methods were evaluated on a range of tasks to assess how each performs across different calibration data. Each pruning method was applied iteratively, with models being pruned and evaluated to ensure that performance was maintained across tasks.

### 3.2 DATASET, TASK AND EVALUATION

To investigate the effects of pruning methods on model generalization, we extensively evaluated three pruning methods. Our experiments were conducted on a set of 14 large language models, including popular models such as OPT-6.7B, LLaMA-2-7B,DeepSeek-R1-Qwen2-7B Guo et al. (2024),Vicuna-7B Chiang et al. (2023),Mistral-7B Jiang et al. (2023), across 24 distinct datasets. Our evaluated datasets cover a diverse set of tasks and are grouped into four primary categories.

- **Sentiment Classification**: SST2 Socher et al. (2013), Yelp, IMDB Rothe et al. (2018), Sentiment140 Go et al. (2009). The tasks were focused on classifying text based on the sentiment it expresses.

- **Question Answering**: ARC-C Clark et al. (2018), ARC-E, OpenBookQA Mihaylov et al. (2018), PubMedQA Jin et al. (2019), RACE Lai et al. (2017), BoolQ Clark et al. (2019), QNLI Wang (2018). The tasks provided answers to specific questions based on provided context.

- **Semantic Similarity**: MRPC, QQP, WIC, SciTail Khot et al. (2018), RTE, MedNLI Romanov & Shivade, PAWS Yang et al. (2019). The tasks assessed the similarity between different pieces of text.

- **Logical Reasoning**: COPA Afshar et al. (2018), LogiQA Liu et al. (2020), SciQ Welbl et al. (2017), WNLI, WinoGrande Sakaguchi et al. (2021), WSC. The tasks required the model to reason logically and solve problems based on textual input.

Among these, sentiment classification and semantic similarity tasks are particularly sensitive to subtle semantic shifts. Sentiment classification relies on detecting diverse emotional cues, while semantic similarity requires deeper understanding of contextual and relational meaning. This motivates our decision to include them in our study of pruning effects.

The calibration data used in these experiments includes: C4, WikiText2, PTB, ARC-C Clark et al. (2018), ARC-E Clark et al. (2018), BoolQ, RTE, SST2, and WNLI. These data sets represented a diverse set of domains and tasks to ensure the robustness of the findings in various use cases. These various datasets were used to well explore the impact of different calibration data on pruning effectiveness.

The performance of each model was evaluated using the standard metrics, accuracy. Our evaluation process includes: (1) Model pruning: This involved setting different calibration data, sentence lengths, and sparsity ratios for the dense models. (2) Evaluation: After pruning, we tested the models on 24 different datasets.

By evaluating these various tasks and datasets, we gain a comprehensive understanding of the generalizability of the pruning methods.

## 3.3 NEURON SEMANTIC ATTRIBUTION

During our evaluations, we find it is fundamentally challenging to understand the unexpected performance variance of pruning methods on some tasks with different calibration data. To our knowledge, the explainability of pruning methods in the aforementioned behaviors is greatly overlooked by the society. To bridge this gap, we propose a new method, Neuron Semantic Attribution (NSA). NSA can learn the relationship between neurons and tokens in the input text, allowing us to build the links between neurons and specific tokens before and after pruning.

The NSA method includes three main steps: *semantic clustering*, *neuron matching*, and *comparison*. First, we identify influential words that are important for specific tasks. Next, we match neurons in the model to these selected words by analyzing their activations. Finally, we compare neuron activations between the unpruned and pruned models to understand the impact of pruning on task-relevant neurons. For example, if the activation related to certain semantics decreases significantly after pruning, it suggests that pruning has removed neurons important for those semantics, potentially explaining performance degradation such as shown in Figure 1.

These steps are detailed as below:

**Step 1: Words Selection** To understand which parts of the input text are important for a specific task, we group the tokens in each input into two categories: those that are relevant to the task and those that are not. This grouping is based on how much each token contributes to the model's decision.

Instead of simply picking keywords, we treat this as a clustering process guided by task relevance. This process is assisted by a large language model, which helps group input tokens into task-relevant and task-irrelevant clusters, as illustrated in Figure 1. For example, in sentiment classification, words like "unnecessary" or "sour taste" are grouped as task-relevant because they clearly express opinion or emotion. These relevant tokens are then used in the next step to link specific neurons to meaningful input features.

This process helps us focus on the parts of the input that matter most, making it easier to analyze how pruning affects the model's ability to handle important information.

| Task | | Calibration data | | | | | | | | |
|---|---|---|---|---|---|---|---|---|---|---|
| | | ARC-C | ARC-E | BoolQ | C4 | PTB | RTE | SST2 | WikiText2 | WNLI |
| Sentiment Classification | Yelp | 70.73 | 72.15 | 73.99 | **75.03** | 73.03 | 71.83 | 71.30 | 73.50 | 68.77 |
| | IMDB | 61.99 | 60.99 | 62.91 | **66.98** | 62.32 | 62.41 | 60.71 | 68.58 | 67.55 |
| | Sentiment140 | 56.49 | 57.35 | 57.60 | 57.59 | **57.73** | 56.47 | 53.93 | 57.57 | 56.07 |
| | SST2 | 66.19 | **68.80** | 67.22 | 66.95 | 67.07 | 68.53 | 64.54 | 67.88 | 66.60 |
| Question Answering | ARC-C | 33.22 | 32.92 | 31.93 | 33.23 | 32.92 | 31.22 | 30.45 | **33.41** | 32.11 |
| | ARC-E | 65.66 | 65.00 | 64.24 | 65.48 | 64.63 | 62.52 | 62.33 | **66.22** | 62.83 |
| | OpenBookQA | 26.17 | 25.95 | 25.75 | **26.51** | 25.59 | 25.13 | 23.73 | 26.47 | 23.65 |
| | PubMedQA | 64.00 | 64.05 | 63.31 | 64.28 | **65.49** | 63.76 | 63.65 | 63.88 | 62.57 |
| | BoolQ | 70.49 | 69.51 | 70.52 | **71.81** | 70.76 | 69.49 | 68.94 | 70.47 | 68.34 |
| | QNLI | **50.92** | 50.69 | 50.66 | 50.33 | 50.60 | 50.65 | 50.84 | 50.16 | 50.85 |
| Semantic Similarity | MRPC | 57.58 | 61.35 | 59.17 | 62.28 | 61.74 | **62.58** | 57.80 | 61.63 | 60.50 |
| | QQP | **47.49** | 45.11 | 44.70 | 44.19 | 44.26 | 44.79 | 44.43 | 45.21 | 43.59 |
| | SciTail | 43.77 | 41.59 | 44.74 | 42.66 | 42.84 | **45.63** | 42.98 | 44.02 | 43.98 |
| | WIC | 50.06 | **50.30** | 49.90 | 49.78 | 50.15 | 50.24 | 50.02 | 50.12 | 50.11 |
| | MedNLI | **34.88** | 34.78 | 34.68 | 34.71 | 34.33 | 34.60 | 34.49 | 34.78 | 34.68 |
| | RTE | 58.53 | 57.70 | **59.01** | 58.52 | 58.74 | 57.94 | 57.87 | 58.70 | 57.56 |
| | PAWS | 46.90 | 46.82 | 46.68 | 46.32 | 46.58 | **47.46** | 47.07 | 46.95 | 46.03 |
| Logical Reasoning | COPA | 79.77 | 80.06 | 79.36 | **81.13** | 80.00 | 80.15 | 78.30 | 81.01 | 79.20 |
| | LogiQA | 23.32 | **23.76** | 23.43 | 23.67 | 23.22 | 23.36 | 22.99 | 23.42 | 23.48 |
| | SciQ | 90.63 | 90.57 | 90.53 | 91.17 | **91.26** | 90.38 | 90.20 | 90.63 | 89.80 |
| | WinoGrande | 63.64 | 63.00 | 63.93 | **64.98** | 63.95 | 63.37 | 62.59 | 64.33 | 63.71 |
| | WNLI | 47.06 | 46.57 | 46.53 | 46.80 | 47.08 | 46.44 | 46.42 | 46.45 | **47.27** |
| | WSC273 | 77.31 | 75.86 | 77.96 | 78.63 | **78.74** | 77.63 | 76.38 | 78.66 | 77.79 |
| | RACE | 37.50 | 37.16 | 37.38 | **38.41** | 38.34 | 37.35 | 37.23 | 38.21 | 36.59 |

Table 1: Accuracy of different calibration data and tasks averaged over 3 pruning methods and 2 sparsities {0.25, 05}.

**Step 2: Neuron Matching** Once the task-relevant tokens have been clustered in Step 1, we input the original text into the unpruned model and record the activations of all neurons across the input tokens.

There are two ways to perform neuron matching. The first is to directly analyze activations from the original model. The second approach enhances interpretability by inserting a Sparse Autoencoder (SAE) after each MLP layer. In this way, the SAE maps dense neuron activations to a sparse space, helping to disentangle overlapping semantic signals and achieve better interpretability as illustrated in Figure 1. The details of the architectures of SAE and its implementation can be found in Section A.3.

We use a score function to quantify each neuron's relevance, which can be replaced by other functions; in our experiments, the score is defined as: $Score = \frac{\sum_{m \in S} A_m}{\sum_{n=1}^{N} A_n}$, where $S$ is the set of influential tokens, $A_m$ is the activation for token $m$, and $N$ is total tokens. Neurons with top relevance scores are deemed task-relevant.

Finally, we associate each task-relevant token with the top-scoring neurons, allowing us to interpret how specific neurons respond to meaningful semantic cues. This matching process is flexible and can be extended with alternative scoring strategies within the NSA framework.

**Step 3: Comparison of the Unpruned and Pruned** We then feed the same input text into the pruned model and calculate activations of the previously selected neurons on the influential tokens. By comparing these activation maps before and after pruning, we identify neurons whose activations decrease significantly. Such decreases indicate that pruning has removed neurons highly related to specific semantics, which potentially leads to observed performance degradation.

**Discussions** NSA is designed as a flexible framework. It allows for various techniques to select task-relevant tokens. NSA is compatible with different large language models, such as LLaMA and OPT, and gives users the freedom to choose which data samples they want to examine. This method enables per-sample visualization and interpretation, offering practical insights into how pruned models behave and where their weaknesses might lie. It also helps pave the way for future research on the effects of pruning on large language models, providing a solid foundation for understanding pruning's impact on model performance.

By looking at how neuron activations relate to model performance, we can find out if pruning introduces unwanted biases or reduces performance on certain tasks. This allows us to make better decisions about pruning strategies and provides valuable insights into the trade-offs involved when simplifying large language models.

## 3.4 ACTIVATION PRUNING

To examine how activations influence pruning (in the spirit of NSA), we fine-tune the pretrained model on a small C4 subset ($|\mathcal{D}_s|{=}400$) with a masked L2 penalty that only acts on low-activation neurons. Let $S$ be the set of channels/neurons whose average activation ranks in the bottom 20% over $\mathcal{D}_s$, and let $M_S$ be a binary mask that is 1 on weights incident to $S$ and 0 elsewhere. We optimize

$$\bar{\mathcal{L}}_\rho(W) \;=\; \mathcal{L}_{\text{task}}(W) \;+\; \frac{\rho}{2}\left\| M_S \odot W \right\|_F^2, \tag{1}$$

then derive a structured 2:4 mask from the magnitude of the *penalized* weights, but finally apply that mask to the *original* $W_0$ to preserve weight values where kept (structural stability).

---

**Algorithm 1** Activation-Guided 2:4 Pruning with Masked L2

---

**Input:** Pretrained weights $W_0$; sample set $\mathcal{D}_s \subset$ C4 with $|\mathcal{D}_s|{=}400$; penalty $\rho$; percentile $p{=}20\%$;
  steps $T$; optimizer $\mathcal{O}$
1: $W \leftarrow W_0$
2: **for** each layer $\ell$ **do**
3:     Run forward passes on $\mathcal{D}_s$; compute per-channel stats $a_\ell[c] \leftarrow \mathbb{E}_{(x,y)\in\mathcal{D}_s}\big[\|h_\ell^{(c)}(x)\|_1\big]$
4:     $S_\ell \leftarrow$ indices of bottom-$p$ channels by $a_\ell$
5:     Build binary mask $M_\ell$ that is 1 for weights incident to $S_\ell$, else 0
6: **for** $t = 1$ to $T$ **do**                                             ▷ masked-L2 fine-tuning on $\mathcal{D}_s$
7:     Sample minibatch $\mathcal{B} \subset \mathcal{D}_s$
8:     $\mathcal{L} \leftarrow \mathcal{L}_{\text{task}}(W;\mathcal{B}) \;+\; \frac{\rho}{2}\sum_\ell \|M_\ell \odot W_\ell\|_F^2$
9:     $W \leftarrow \mathcal{O}\big(W, \nabla_W \mathcal{L}\big)$
10: Construct mask $P$ by selecting top-$S$% entries in $|W|$, setting them to 1 and others to 0
11: **Return** pruned model $\tilde{W} \leftarrow P \odot W_0$                 ▷ apply mask to original weights

---

## 4 EXPERIMENTS

**Calibration Data** Calibration data is important for pruning methods. Table 1 displays the accuracy from 24 datasets/tasks (e.g. ARC-E) and 9 calibration datasets (e.g. ARC-C) averaged over 3 pruning methods and 2 sparsities $\{0.25, 0.5\}$. In addition, Table 2 presents the accuracy of 14 pruned models (including the very new DeepSeek-R1-Qwen2-7B Guo et al. (2024)) using the same pruning method Wanda with different calibration data.

**Insensitive to Calibration Data** From Table 1, we can see the performance of Semantic Similarity (e.g., MRPC and QQP) and Logical Reasoning (e.g., SciQ and WNLI) is relatively robust to calibration data. In particular, the performance of SciQ over different calibration sets are very stable and consistent. We believe these tasks likely depend on more general patterns that are less sensitive to variations in calibration data. These findings indicate that pruning methods can maintain performance on these tasks irrespective of the calibration data.

**Sensitive to Calibration Data** From Table 1, Sentiment Classification like Yelp and SST2, and Question Answering like ARC-E, show significant performance variability depending on the calibration data. From Table 2, in particular, we can clearly see the significant performance variability. For example, the accuracy of the opt-6.7B model on the Yelp dataset reaches 74% with the Wiki-Text2 calibration data, compared to approximately 60% with other datasets. Similarly, for the ARC-E task, the DeepSeek-R1-Qwen2-7B model shows an accuracy of 66.58% vs. 35.69%. with the C4 and SST2 calibration data respectively. Calibration data plays a crucial role for the tasks SST2, Yelp, and ARC-E, where different calibration data, such as WikiText2, can lead to a significant improvement in model performance.

In conclusion, this analysis demonstrates that selecting the appropriate calibration data is crucial for optimizing pruning performance. For sentiment classification and question answering, dataset selection can have a significant impact on model performance. Our findings underscore the importance of understanding the role that calibration data play in pruning LLMs and emphasize the need for further exploration into optimizing calibration data for different tasks.

| Model | Task | Calibration Data | | | | | | | | |
|---|---|---|---|---|---|---|---|---|---|---|
| | | ARC-C | ARC-E | BoolQ | C4 | PTB | RTE | SST2 | WikiText2 | WNLI |
| DeepSeek-R1-Qwen2-7B | Yelp | 50.04 | 50.04 | 60.53 | **89.90** | 82.66 | 50.91 | 50.11 | 87.79 | 50.08 |
| LLaMA-3-8B | Yelp | 75.51 | 76.80 | 50.37 | 95.60 | **96.90** | 73.64 | 50.00 | 95.85 | 59.79 |
| LLaMA-7B | Yelp | 55.35 | **63.94** | 57.31 | 50.38 | 50.01 | 61.15 | 53.53 | 50.24 | 57.87 |
| OPT-13B | Yelp | 65.57 | 61.96 | 53.18 | **70.89** | 55.92 | 53.57 | 63.28 | **70.89** | 68.04 |
| OPT-30B | Yelp | 83.54 | 74.96 | 73.27 | 71.73 | 71.28 | 79.22 | 74.54 | 61.36 | **86.14** |
| OPT-6.7B | Yelp | 53.65 | 55.38 | 67.43 | 66.37 | 59.47 | 56.54 | 61.17 | **74.18** | 52.46 |
| Vicuna-7B | Yelp | 88.23 | 85.02 | 80.25 | 95.22 | 92.22 | 91.37 | 53.46 | **95.47** | 89.14 |
| DeepSeek-R1-Qwen2-7B | ARC-E | 48.86 | 47.47 | 54.84 | 66.58 | 64.02 | 49.71 | 35.69 | 66.08 | 42.17 |
| LLaMA-3-8B | ARC-E | 62.37 | 60.98 | 62.84 | 70.08 | 69.91 | 59.18 | 40.57 | **70.66** | 53.79 |
| LLaMA-7B | ARC-E | 62.04 | 61.03 | 61.87 | **69.70** | 68.18 | 60.86 | 56.02 | 69.65 | 57.49 |
| OPT-13B | ARC-E | 61.28 | 61.15 | 60.77 | **63.01** | 57.53 | 58.12 | 51.98 | 62.21 | 52.10 |
| OPT-30B | ARC-E | 63.13 | 62.96 | 64.60 | **68.06** | 64.18 | 61.66 | 51.81 | 66.75 | 52.69 |
| OPT-6.7B | ARC-E | 49.28 | 49.62 | 50.88 | **57.45** | 53.75 | 49.45 | 50.00 | 56.40 | 45.66 |
| Vicuna-7B | ARC-E | 64.65 | 65.57 | 66.20 | 69.23 | **70.16** | 64.69 | 60.98 | 69.23 | 62.37 |

Table 2: Accuracy of pruned models across different calibration data and evaluation on Yelp and ARC-E dataset using Wanda pruning.

### 4.1 NEURON SEMANTIC ATTRIBUTION (NSA)

**Motivation** For sentiment classification like Yelp in Table 2, we observe a noticeable performance variance in certain models after pruning. This variance in performance prompted us to explore the underlying reasons behind it. The core of this exploration is to understand what has been pruned—which components of the model are being removed, and how this removal affects its ability to perform sentiment classification. To investigate this, we applied our analysis method, Neuron Semantic Attribution (NSA), to analyze the relationship between tokens and the corresponding neuron activations. Specifically, we sampled texts from the Yelp dataset. We visualized how the tokens are connected to the neurons in the model.

**NSA Visualization on Yelp Data** In Figure 1, we show an analysis example of this NSA method. The depth of the color represents the activation strength of the neurons connected to the tokens. We observe that the color intensity of the highlighted words changes significantly between the unpruned and pruned models. In addition, we can see that the activation of key sentiment-related tokens, such as 'unnecessary' showed a decrease in activation strength after pruning. This suggests that the pruning weakens of the neurons responsible for sentiment analysis, explaining the performance decline.

Interestingly, although there is a decrease in activation for the pruned model, some influential tokens still retain a significant level of activation, which explains why the pruned models can still maintain reasonably good performance for sentiment classification. However, the reduced activation strength may indicate that the model is less confident in its decision-making process, potentially contributing to the observed performance gap.

More NSA visualizations for other tasks are included in the section A.2.

### 4.2 ACTIVATION PRUNING

We study how activation-aware regularization affects semi-structured pruning. Following Sec. 3.4, we fine-tune the pretrained model on a 400-example C4 subset with a *masked* $\ell_2$ penalty that only acts on the bottom-20% activation channels, then derive a 2:4 mask from the penalized weights and finally apply that mask to the original parameters for structural stability. Formally, $\bar{\mathcal{L}}_\rho(W) = \mathcal{L}_{\text{task}}(W) + \frac{\rho}{2} \|M_S \odot W\|_F^2$, where $S$ indexes low-activation channels and $M_S$ is the corresponding binary mask.

**Setup.** We sweep the regularization strength $\rho \in \{10^{-1}, 10^{-2}, 10^{-3}, 10^{-4}, 10^{-5}, 0\}$ on LLaMA-2-7B and Qwen2.5-VL-7B.

**Findings.** Strong penalties cause divergence; small–moderate penalties are stable with model-dependent optima (LLaMA-2-7B peaks at $10^{-3}$, Qwen2.5-VL-7B at $10^{-4}$). In short, gentle activation-aware regularization helps, while over-penalization collapses useful low-activation pathways; set $\rho$ conservatively and prefer diverse calibration data to preserve neuron coverage.

| Model | $\rho=10^{-1}$ | $10^{-2}$ | $10^{-3}$ | $10^{-4}$ | $10^{-5}$ | 0 |
|---|---|---|---|---|---|---|
| LLaMA-2-7B | inf | inf | 10.32 | 10.92 | 10.96 | 10.40 |
| Qwen2.5-VL-7B | inf | inf | 10.48 | 9.89 | 13.67 | 15.82 |

Table 3: **Activation pruning sweep over $\rho$.** "inf" indicates training divergence under too-strong penalty.

## 5 DISCUSSION

**Calibration and Evaluation Data** As shown in Table 5, pruning performance is sensitive to the choice of calibration data. Interestingly, pruning with the same dataset as the target task does not necessarily yield the best results. For instance, when evaluating on SST2, using SST2 itself as calibration data only achieves 64.54% accuracy, while using ARC-E leads to 68.80%. Similarly, for the ARC-C task, WikiText2 slightly outperforms ARC-C as calibration. These observations contradict common practices in model training, where using task-specific data typically leads to better performance, inspiring more explorations in this field in the future.

**Toward Constructing Reliable Calibration Sets** Given the significant impact of calibration data, an open question is how to systematically select or construct calibration sets that generalize well across tasks. Our empirical results suggest that datasets with broad semantic coverage (e.g., C4, WikiText2) serve as robust calibration candidates. However, for tasks like semantic similarity or question answering, tailored calibration sets aligned with task-specific semantics may still offer advantages. Future research could explore automatic metrics for estimating semantic coverage and task alignment to guide calibration set construction.

**Activation Changes in Tokens** As shown in Figure 1, pruning affects the model's attention to task-relevant tokens in different ways. For example, tokens such as "price" and "unnecessary" show a clear drop in neuron activation after pruning, indicating that the model becomes less focused on these key tokens. On the other hand, tokens like "bedside" and "sour taste" exhibit stable or even increased activations, suggesting that the model still attends to them. These variations in activation among task-relevant tokens help explain the observed performance differences after pruning.

We also observe activation changes in tokens that are not directly related to the task. However, these changes do not directly explain performance shifts, since the model's predictions should not depend on these tokens. Furthermore, the neurons visualized in our analysis are selected based on their relevance to task-specific tokens.

**Why Sentiment Classification Is Particularly Sensitive** As discussed earlier, sentiment classification appears to be more sensitive to pruning compared to other tasks. One possible reason is the instability in how sentiment-relevant tokens are represented in the model. For instance, in Figure 1, tokens such as "berating", "atrocious", and "unnecessary" show substantial changes in activation after pruning. Apparently, these words are highly relevant to sentiment classification. In contrast, these words do not affect the performance of other tasks significantly. Additionally, we find that the activation of these sentiment-bearing tokens is strongly influenced by the choice of calibration data, further contributing to the variability in performance.

## 6 CONCLUSION

In this paper, we present a comprehensive analysis of pruning generalizability across diverse tasks, calibration datasets, and pruning methods, revealing that pruned models exhibit task-specific performance sensitivities. To better understand these differences, we introduce Neuron Semantic Attribution (NSA), a novel explainability method tailored for pruning analysis. NSA assigns interpretable semantics to neurons by linking them to task-relevant input features, enabling us to visualize how pruning affects model behavior at a fine-grained level. Unlike existing interpretability studies, NSA is the first to focus on explaining pruning changes, providing a reliable method to evaluate and guide pruning strategies from an interpretability perspective.

ETHICS STATEMENT

This work investigates pruning methods for large language models from the perspective of neuron semantic attribution. All experiments were conducted on publicly available pretrained models and benchmark datasets, without the use of private or personally identifiable data. The proposed method aims to improve model efficiency and interpretability, thereby reducing computational cost and energy consumption associated with large-scale model deployment. We do not foresee direct societal risks from this research, but acknowledge that any pruning technique could potentially be misused to compress harmful models for wider distribution. Our study complies with the ICLR Code of Ethics and is intended to support responsible and sustainable AI development.

REPRODUCIBILITY STATEMENT

We have made every effort to ensure the reproducibility of our work. All pruning algorithms, neuron semantic attribution procedures, and experimental setups are described in detail in Sections 3 and 4. Additional implementation details, hyperparameter configurations, and extended experimental results are provided in the Appendix A.3. We rely exclusively on publicly available pretrained models and benchmark datasets, with preprocessing steps documented in the supplementary materials. To further facilitate reproducibility, we will release anonymous source code and scripts as part of the supplementary submission. Theoretical results and proofs are included in the appendix to ensure the clarity and verifiability of the claims.

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

# A   APPENDIX

## LLM USAGE STATEMENT

Large language models (LLMs) were used as general-purpose assistive tools during the preparation of this paper. Specifically, they were employed to improve the fluency of English writing, polish grammar, and provide suggestions for phrasing, without generating novel scientific ideas or experimental results. All research design, algorithm development, experiments, and analyses were conducted entirely by the authors. The authors take full responsibility for the content of this work, and no LLM is considered as an author or contributor.

## A.1   COMPONENT ANALYSIS

**Downstream Tasks** We also conducted a thorough analysis of the performance differences across four downstream tasks. In Figure 3, our results show that Sentiment Classification tasks exhibit significant performance variations across different calibration data, demonstrating a high degree of sensitivity. The performance variations are explained by our NSA visualization in Section 4.1. In contrast, Question Answering and Logical Reasoning tasks show relatively stable performance. Specifically, for tasks such as those in the ARC-C and ARC-E datasets, the model's performance does not exhibit significant variations with changes in calibration data. This suggests that these types of tasks rely more on reasoning processes. These tasks generally involve extracting key information from provided contexts, and this reasoning ability depends less on the precise composition of the calibration data, making them more robust to such changes.

**Cross-Model Pruning** It is also interesting to know how each LLM performs over different calibration data. Table 5 shows the impact of different calibration data on pruning results for each model. For most models, datasets like C4 and WikiText2 consistently lead to higher performance after pruning, suggesting they are more effective at preserving model capacity. For instance, LLaMA-13B performs best on WikiText2 (61.27%) and RTE (61.25%), while vicuna-13B achieves its highest score on C4 (61.06%). Conversely, datasets such as WNLI and ARC-C often result in lower performance, indicating their limited effectiveness for pruning calibration data.

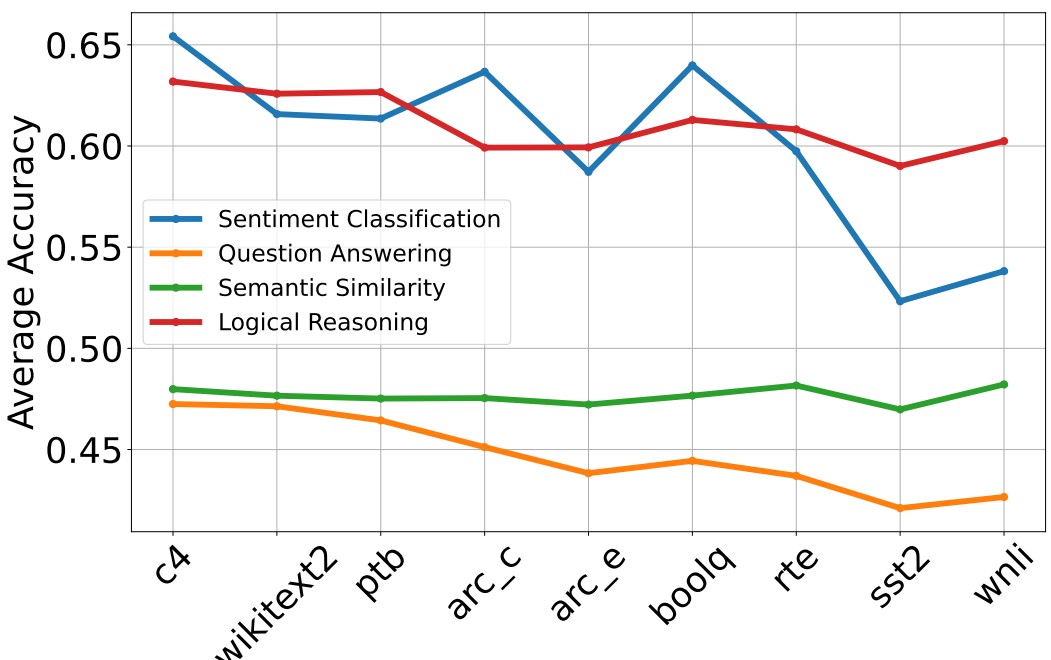

Figure 3: Accuracy of pruned models on different calibration data with 2:4 sparsity averaged over 4 kinds of tasks.

|  | C4 | WikiText2 | PTB | ARC-C | ARC-E | BoolQ | RTE | SST2 | WNLI |
|---|---|---|---|---|---|---|---|---|---|
| Coverage | 95.3 | **96.8** | 84.5 | 85.5 | 85.8 | 94.2 | 96.2 | 83.8 | 86.9 |
| Avg. Accuracy | **55.96** | 54.74 | 54.49 | 54.06 | 52.43 | 54.34 | 53.11 | 50.11 | 51.23 |

Table 4: Semantic coverage and average accuracy of pruned models for different calibration data.

Overall, the impact of calibration data varies across models. Larger models like LLaMA-13B and Vicuna-13B exhibit relatively stable performance across datasets, while smaller models like Mistral-7B and OPT-6.7B show greater variability, reflecting a stronger dependence on the choice of calibration data. This suggests that selecting appropriate datasets, such as C4 or WikiText2, can improve pruning outcomes, especially for smaller models.

**Semantic Coverage and Pruning Effectiveness** To understand what makes a calibration dataset effective for pruning, we measure its *semantic coverage*—defined as one minus the average cosine similarity among sentence embeddings using the all-MiniLM-L6-v2 model. A higher coverage indicates more semantic diversity within the dataset. The resulting metric, referred to as *semantic coverage*, is shown in Table 4.

Interestingly, we observe a strong correlation between semantic coverage and pruning effectiveness, measured by the average accuracy of pruned models across evaluation tasks. Datasets with high semantic coverage(95.3% and 96.8%), such as C4 and WikiText2, yield the best average accuracies (55.96% and 54.74%). In contrast, datasets like SST2 and WNLI, with lower coverage (83.8% and 86.9%), result in weaker pruning performance (50.11% and 51.23%).

These results suggest that semantically diverse calibration data activates a broader range of neurons during pruning, helping preserve more informative parameters. In contrast, homogeneous datasets may bias pruning toward limited representations, reducing generalization. Therefore, semantic diversity within calibration data is a critical factor for effective and robust pruning. We encourage future pruning studies to incorporate semantic diversity estimation into the calibration data selection process.

More experiment details and analysis are included in the sections A.3 and A.4.

| Model | Calibration Data | | | | | | | | |
|---|---|---|---|---|---|---|---|---|---|
| | ARC-C | ARC-E | BoolQ | C4 | PTB | RTE | SST2 | WikiText2 | WNLI |
| DeepSeek-R1-Qwen2-7B | 49.12 | 49.12 | 49.22 | **53.10** | **53.10** | 48.81 | 48.61 | 52.50 | 48.45 |
| LLaMA-13B | 59.55 | 60.42 | 60.78 | 60.55 | 60.49 | 61.25 | 58.75 | **61.27** | 58.88 |
| LLaMA-3-8B | 55.91 | 55.86 | 55.95 | **57.29** | 57.19 | 55.65 | 54.40 | 57.19 | 55.33 |
| LLaMA-7B | 52.54 | 52.01 | 52.09 | 52.55 | **53.26** | 51.86 | 51.99 | 52.46 | 51.99 |
| Mistral-7B | 53.45 | 53.46 | 53.37 | **54.62** | 54.42 | 52.86 | 52.52 | 54.33 | 52.77 |
| OPT-13B | 54.11 | 53.53 | 53.92 | **54.35** | 53.47 | 53.78 | 54.27 | 53.33 | 53.80 |
| OPT-30B | 53.91 | 53.58 | 53.95 | **54.73** | 54.03 | 53.81 | 52.53 | 54.41 | 53.33 |
| OPT-6.7B | 52.82 | 52.96 | 53.28 | **54.05** | 53.14 | 52.31 | 52.27 | 53.64 | 52.13 |
| Vicuna-13B | 59.00 | 59.00 | 59.56 | **61.06** | 60.62 | 58.73 | 58.41 | 60.79 | 57.98 |
| Vicuna-7B | 58.34 | 58.27 | 58.17 | 58.78 | **59.00** | 57.35 | 56.60 | 58.69 | 57.13 |

Table 5: Accuracy of pruned models on different calibration data with 2:4 sparsity averaged over 3 pruning methods and 24.

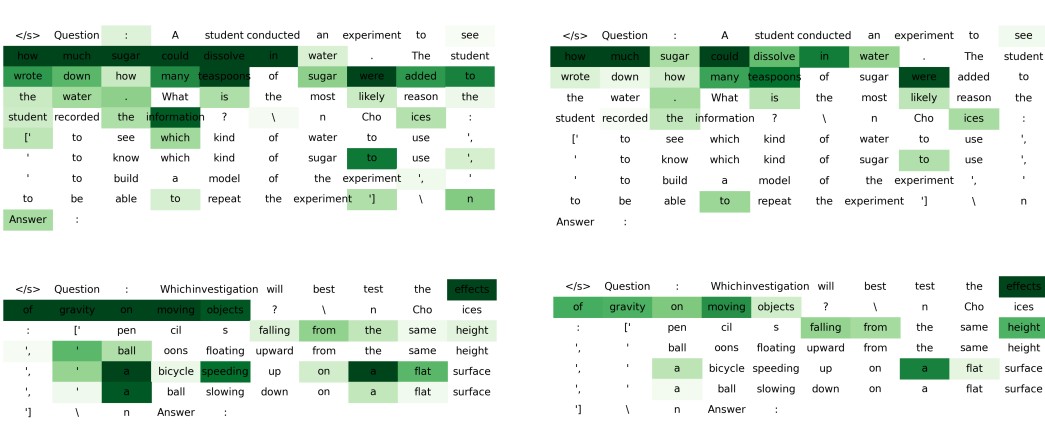

**Unpruned OPT-6.7B**      **Pruned with RIA method C4 data (2:4 sparsity)**

Figure 4: The activation of neuron 8217 and neuron 8283 in layer 23.The pruned model is OPT-6.7B with RIA method and calibration dataset C4. Deeper colors in the visualization represent stronger neuron activations for specific tokens.

## A.2 NSA VISUALIZATION

**NSA Visualization on ARC-C Data** We also applied NSA to the ARC-C dataset, as shown in Fig.4, which presented a different result.

In the case of the ARC-C dataset, the results were quite different. The visualization in Fig.4 reveals that, after pruning, the model retains a strong focus on key tokens related to the task, such as words representing question cues or common reasoning patterns. Unlike sentiment classification, where pruning caused a noticeable reduction in activation strength, the impact on the ARC-C task was less pronounced. The tokens connected to reasoning and question-answering were still highly activated post-pruning, suggesting that the model maintained its ability to reason through the problem.

In fact, the intensity of activation in ARC-C remained relatively stable across both the original and pruned models. This suggests that, unlike sentiment classification, tasks like ARC-C may be less sensitive to pruning, possibly due to the inherent structure of reasoning tasks. In this dataset, pruning appears to have less impact on the model's ability to focus on critical decision-making tokens, which might explain why the performance drop was less significant compared to tasks like sentiment classification.

**NSA Visualization on IMDB Data** Figure 5 presents the activation of neuron 2287 in layer 23 of the pruned model (LLaMA-3-8B) using the Wanda method and the PTB calibration dataset.

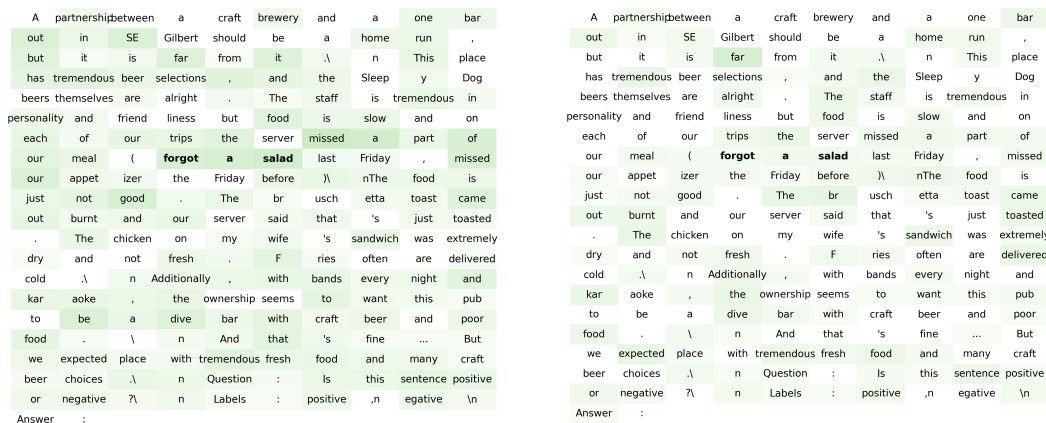

**Unpruned LLaMA-3-8B**          **Pruned with Wanda method PTB data (2:4 sparsity)**

Figure 5: The activation of neuron 2287 in layer 23.The pruned model is LLaMA-3-8B with Wanda method and calibration dataset PTB. Deeper colors in the visualization represent stronger neuron activations for specific tokens.

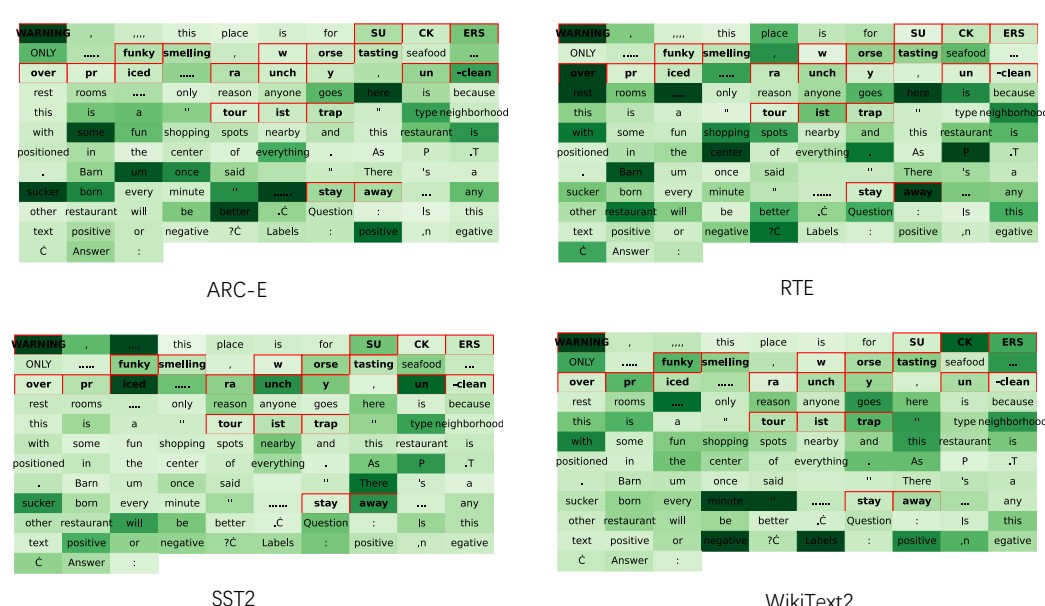

ARC-E          RTE

SST2          WikiText2

Figure 6: The activation of neuron 47 in layer 23.The pruned model is LLaMA-3-8B with RIA method and various calibration dataset. Deeper colors in the visualization represent stronger neuron activations for specific tokens.

**NSA Visualization on Yelp Data with Different Calibration Data** Figure 6 presents the activation of neuron 47 in layer 23 of the the pruned model (LLaMA-3-8B) using the RIA method and various calibration dataset.

## A.3 EXPERIMENT DETAILS

**Pruning Type** In our experiments, we evaluate four representative pruning types to comprehensively assess their impact on model performance. Specifically, we include two structured pruning configurations: **2:4 sparsity** and **4:8 sparsity**, where only 2 or 4 weights are retained in every block of 4 or 8 consecutive weights, respectively. These patterns conform to hardware-friendly sparse formats and are widely adopted in practical deployment scenarios. In addition, we consider two unstruc-

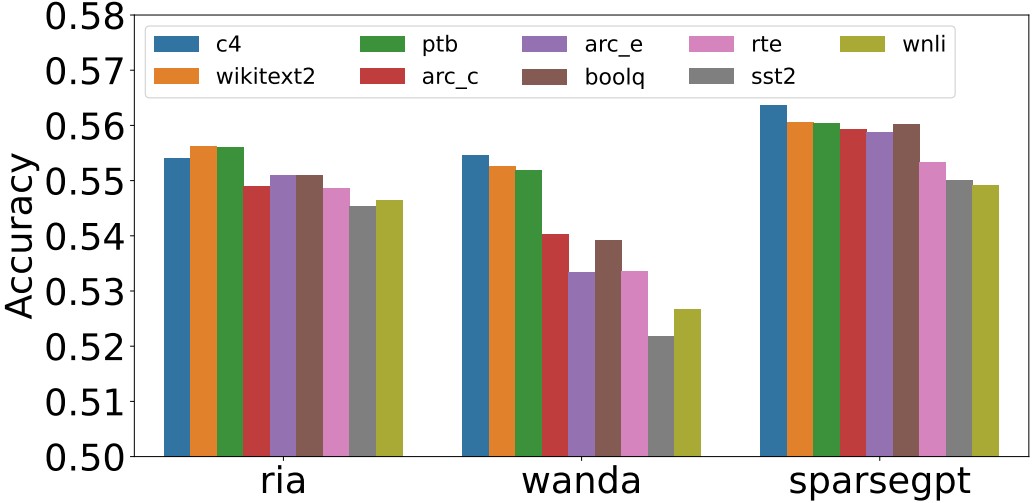

Figure 7: Accuracy of three pruning methods using 9 different calibration data averaged over 24 tasks.

tured pruning settings with global sparsity levels of **25%** and **50%**, where individual weights are removed based on magnitude regardless of their position. These settings allow us to investigate both fine-grained and coarse-grained sparsity effects on model robustness and neuron attribution.

**Details of SAE** To support neuron-level semantic analysis in NSA, we adopt the pretrained SAE checkpoint released by EleutherAI, trained on the residual stream of LLaMA-3-8B. The model uses data from the RedPajama v2 corpus ( 8.5B tokens) and maps activations into a 32× overcomplete sparse space. It is trained with MSE reconstruction loss and sparsity regularization, using a batch size of 2048 and Adam optimizer. This provides interpretable neuron representations that enable NSA to trace how pruning affects task-relevant neuron behavior.

### A.4   OTHER ANALYSIS

**Pruning Methods** In Figure 7, we compare the accuracy of the three pruning methods using different calibration data averaged over 24 tasks. SparseGPT demonstrates the most competitive performance on most calibration data. These extensive evaluations highlight the importance of empirical evaluation in identifying the most effective method for specific tasks.

**Sequence Length of Tokens** To further investigate the impact of sequence length of tokens on the performance of pruned models, we plotted a bar chart in Figure 8 comparing the three pruning methods. The results suggest that, for the pruned models in this study, sequence length has little impact on performance. More detailed experiments on sequence length can be found in our supplementary materials.

**Neuron-Level Analysis of Pruning** To better understand the impact of pruning at the neuron level, we analyze the activation differences between pruned and unpruned models using Neuron Semantic Attribution (NSA). As illustrated in Figure 9, pruning significantly suppresses the activation of certain task-relevant tokens such as *"prices"* and *"leaves"*. This suggests that pruning removes neurons encoding semantically meaningful information, which could explain the observed performance degradation. Moreover, different pruning methods and calibration data prune different sets of neurons, leading to variability in model behavior.

**Sparsity Ratio** Table 6 presents the accuracy trends of pruned models under varying sparsity ratios across different tasks. We observe that the sensitivity to increasing sparsity differs notably between tasks.

| Task | Sparsity ratio | | | | | | | |
|---|---|---|---|---|---|---|---|---|
| | 0.1 | 0.2 | 0.3 | 0.4 | 0.5 | 0.6 | 0.7 | 0.8 |
| ARC-C | 43.25 | 43.17 | 42.62 | 41.17 | 36.82 | 28.41 | 18.47 | 20.39 |
| ARC-E | 76.43 | 76.31 | 75.76 | 73.91 | 69.88 | 60.71 | 30.89 | 27.30 |
| BoolQ | 77.53 | 77.75 | 76.88 | 74.78 | 72.59 | 65.96 | 49.57 | 37.86 |
| COPA | 87.00 | 88.50 | 86.50 | 85.00 | 84.00 | 75.00 | 65.00 | 62.00 |
| LogiQA | 25.88 | 24.73 | 22.66 | 22.04 | 23.88 | 22.12 | 20.05 | 19.66 |
| MRPC | 69.24 | 68.75 | 65.69 | 62.26 | 54.41 | 55.89 | 49.75 | 31.62 |
| OpenBookQA | 31.30 | 32.50 | 31.70 | 30.60 | 28.30 | 22.90 | 12.40 | 14.20 |
| PubMedQA | 71.50 | 71.20 | 70.30 | 70.60 | 70.00 | 58.80 | 41.40 | 33.80 |
| RACE | 39.66 | 39.47 | 39.95 | 40.19 | 39.19 | 34.50 | 25.45 | 23.01 |
| RTE | 61.73 | 63.18 | 63.36 | 58.84 | 54.69 | 52.89 | 52.71 | 51.98 |
| SciQ | 93.80 | 94.00 | 94.30 | 94.45 | 94.15 | 91.00 | 76.40 | 28.00 |
| WinoGrande | 69.22 | 69.53 | 68.90 | 68.78 | 66.85 | 61.60 | 50.63 | 49.73 |
| WSC273 | 80.59 | 80.95 | 82.23 | 82.78 | 80.59 | 73.44 | 55.13 | 50.91 |

Table 6: Accuracy across tasks at different sparsity ratios. We use LLaMA-7B model, RIA pruning, and C4 calibration data.

| Similarity | C4 | Wikitext2 | PTB | Arc-challenge | Arc-easy | BoolQ | RTE | SST2 |
|---|---|---|---|---|---|---|---|---|
| C4 | 100.0 | 93.6 | 92.3 | 90.2 | 91.2 | 91.2 | 90.3 | 91.5 |
| Wikitext2 | | 100.0 | 92.1 | 90.0 | 90.0 | 91.5 | 90.3 | 91.2 |
| PTB | | | 100.0 | 89.3 | 89.3 | 90.3 | 89.6 | 90.4 |
| Arc-challenge | | | | 100.0 | 89.9 | 89.3 | 88.7 | 89.9 |
| Arc-easy | | | | | 100.0 | 89.4 | 88.7 | 90.0 |
| BoolQ | | | | | | 100.0 | 89.5 | 90.3 |
| RTE | | | | | | | 100.0 | 89.7 |
| SST2 | | | | | | | | 100.0 |

Table 7: The similarity (%) of pruned model on various calibration data with LLaMA-2-7B model.

Some tasks demonstrate high robustness to pruning and maintain stable performance until sparsity becomes extreme. For instance, SciQ maintains exceptionally high accuracy from 93.80% at sparsity 0.1 to 94.45% at sparsity 0.4, with only a mild decline to 91.00% at 0.6. However, the performance drops sharply to 28.00% at sparsity 0.8. Similarly, COPA achieves 87.00% at 0.1 and retains strong performance (84.00%) even at 0.5, only declining to 62.00% at the highest sparsity.

In contrast, several tasks exhibit consistent degradation as sparsity increases. For example, ARC-E drops from 76.43% at sparsity 0.1 to 69.88% at 0.5 and further down to 27.30% at 0.8. BoolQ also shows a progressive decline, going from 77.53% to 72.59% at 0.5 and down to 37.86% at 0.8. OpenBookQA falls from 31.30% at 0.1 to just 14.20% at 0.8, and MRPC decreases from 69.24% to 31.62% over the same range.

Interestingly, a few tasks (e.g., WSC273) show relatively stable accuracy across moderate sparsity levels, with performance peaking at 82.78% at sparsity 0.4 before dropping more noticeably beyond 0.6.

Overall, these results highlight that sparsity impacts tasks differently: some tasks are resilient and only degrade at high pruning levels (e.g., SciQ, COPA), while others degrade steadily or abruptly even under moderate sparsity (e.g., ARC-E, MRPC, OpenBookQA). This suggests that task-dependent pruning strategies may be necessary to preserve performance across diverse applications.

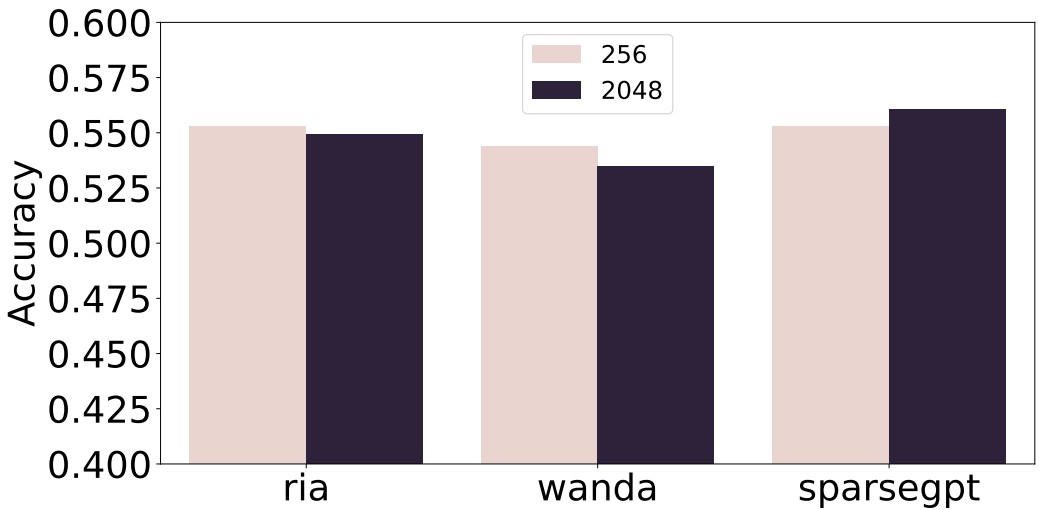

Figure 8: Average accuracy of pruned model with different sequence length of tokens.

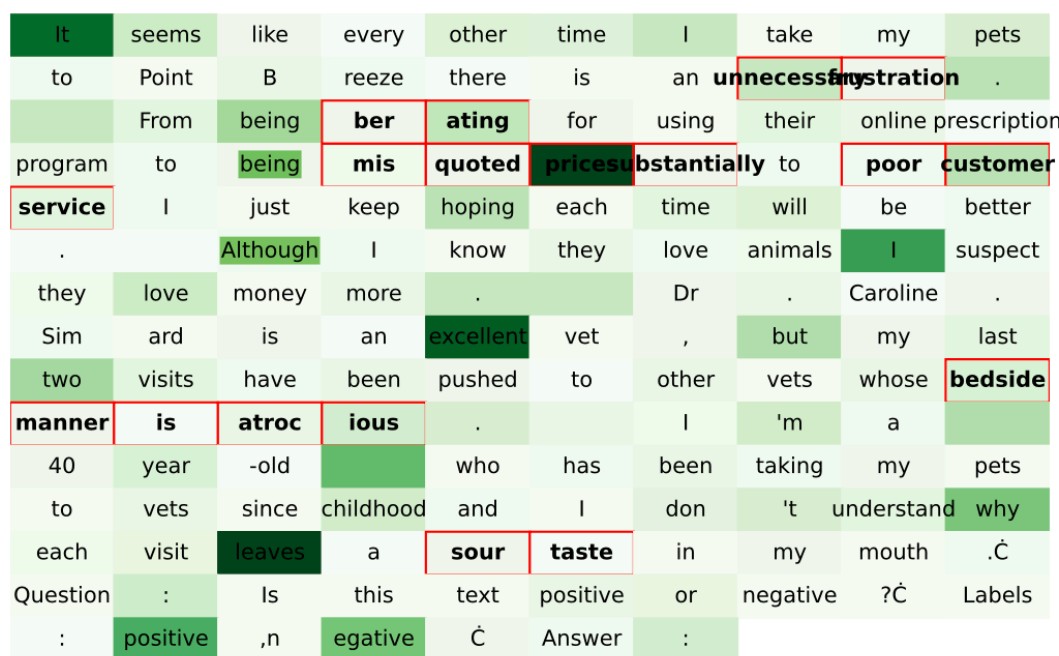

Figure 9: Activation that will be decreased from pruning.

| Seqlen | ARC-C | ARC-E | BoolQ | COPA | IMDB | LogiQA | MedNLI | MRPC | OpenBookQA | PAWS | PubMedQA | QNLI |
|---|---|---|---|---|---|---|---|---|---|---|---|---|
| 64 | **38.57** | **71.17** | 75.23 | 82.00 | **76.75** | 24.12 | 33.19 | 44.12 | 28.00 | 45.77 | 69.60 | 50.85 |
| 128 | 38.23 | 70.96 | 75.35 | **83.00** | 62.70 | 23.96 | 33.33 | 60.05 | 28.60 | 44.36 | **70.20** | 51.33 |
| 256 | 37.63 | 71.13 | **75.90** | 77.00 | 63.88 | 24.73 | 33.33 | 54.90 | **29.00** | 44.72 | 69.40 | **51.84** |
| 512 | 37.12 | 70.41 | 74.40 | 80.00 | 64.70 | 24.73 | 33.33 | 46.08 | 27.00 | **46.17** | 68.60 | 50.92 |
| 1024 | 35.92 | 69.70 | 73.30 | 72.00 | 68.93 | **24.88** | **33.41** | **72.79** | 25.20 | 44.72 | 67.00 | 50.69 |
| 2048 | 32.00 | 63.38 | 57.83 | 73.00 | 66.77 | 21.97 | 33.33 | 35.54 | 21.60 | 44.24 | 64.80 | 50.41 |

| Seqlen | QQP | RACE | RTE | SciQ | SciTail | Sentiment140 | SST2 | WIC | WinoGrande | WNLI | WSC273 | Yelp |
|---|---|---|---|---|---|---|---|---|---|---|---|---|
| 64 | 65.25 | 40.38 | **66.06** | 92.90 | 48.85 | **71.46** | **79.93** | 50.16 | **66.14** | 43.66 | 76.92 | **75.94** |
| 128 | **65.93** | **40.48** | 65.34 | **93.00** | 49.23 | 65.62 | 75.92 | 50.16 | 65.27 | 43.66 | 78.02 | 64.97 |
| 256 | 65.90 | 40.00 | 62.09 | 92.90 | **49.54** | 63.80 | 71.44 | **50.47** | 65.35 | **46.48** | 77.29 | 66.94 |
| 512 | 63.51 | 39.33 | 61.73 | 92.90 | 49.54 | 65.95 | 76.83 | 50.16 | 63.54 | 45.07 | **80.22** | 70.04 |
| 1024 | 63.83 | 38.66 | 62.09 | 92.90 | 48.16 | 68.44 | 76.95 | 50.00 | 62.27 | 42.25 | 78.02 | 74.70 |
| 2048 | 62.98 | 36.65 | 58.48 | 91.10 | 42.56 | 61.18 | 63.19 | 50.00 | 56.67 | 45.07 | 72.89 | 68.08 |

Table 8: Performance across various tasks at different sequence lengths.

**Sequence Lengths of Calibration Data** Table 8 shows that performance varies across different sequence lengths. For tasks like ARC-E and ARC-C, shorter sequences (e.g., 64) yield higher accuracy, while longer sequences (e.g., 2048) tend to decrease performance. Conversely, tasks like MRPC and SciQ benefit from longer sequences, with MRPC's accuracy increasing from 44.12 at seqlen 64 to 72.79 at seqlen 2048. Sentiment tasks like Yelp and SST2 perform best with shorter sequences, suggesting that excessively long sequences might not always be beneficial for certain tasks.

**Similarity Between Pruned Models** We further investigate the parameter-level similarity between models pruned using different calibration data. As shown in Table 7, models pruned with semantically similar calibration data (e.g., C4 and WikiText2) exhibit high similarity scores (¿90%), while those pruned with task-specific datasets tend to diverge more. This indicates that even under the same pruning method, calibration data shapes the final model structure.

