# OpenReview forum: "Revisiting Large Language Model Pruning using Neuron Semantic Attribution"
_ICLR.cc/2026/Conference — Submitted to ICLR 2026_

### Official Review · Reviewer_K4fY · 2025-10-26

**Soundness:** 1
**Presentation:** 2
**Contribution:** 2
**Rating:** 0
**Confidence:** 4

**Summary:**

The paper shows that pruning does not affect all downstream tasks in large language models (LLMs) equally: some tasks remain relatively stable after pruning, while others, for example, sentiment classification experience much larger performance degradation, and this sensitivity depends on the calibration data used for pruning. To understand why this happens, the authors introduce Neuron Semantic Attribution (NSA), an interpretability framework that links task relevant tokens to the neurons that activate for those tokens and then measures how pruning changes those neuron activations. Using NSA, they argue that pruning can disproportionately suppress neurons that encode task-critical semantics (e.g., sentiment-bearing words), which in turn explains the larger accuracy drops seen in vulnerable tasks.

**Strengths:**

The paper evaluates pruning across many language models, multiple sparsity settings, different pruning methods, and a broad set of downstream tasks, showing that their findings are general behaviour rather than something tied to one model or benchmark.

The paper provides insights regarding the role of calibration data used for pruning and shows that it is paramount to downstream performance.

The proposed framework provides a structured way to analyse why model performance degrades after pruning by identifying which task-relevant neurons are being weakened or removed, which in turn could help develop more informed pruning strategies.

**Weaknesses:**

The framework outlined is mostly heuristic and offers limited novelty compared to existing interpretability work:
1) The task specific tokens are extracted using LLMs. These are not causally validated. As a result, the framework assumes the correctness of these token groups without establishing that they truly drive the model’s behaviour on the task.

2) The score function relies on the oversimplified assumption that high activations are important; this assumption has been debunked, as prior interpretability work has shown that neurons with low or indirect activation can still be essential, so this assumption is unreliable.

3) The method does not distinguish globally important / shared neurons (from task-local ones, so drops in activation are overinterpreted as “task damage” without checking whether those neurons are just broadly useful everywhere. Moreover, recent work has shown that there are neurons that activate highly within the model, see massive activations[1].

4) Neuron attribution is already widely studied using approaches such as causal tracing, linear probes, and circuit analysis; in that context, the paper mostly offers a visual diagnostic tool rather than a substantively new interpretability method.


The pruning evaluation averages across different models, sparsity levels, pruning methods, and tasks. This makes it hard to attribute effects to pruning itself rather than model differences. Baseline unpruned performance results are missing.

From Section 4.2, it is unclear what the results indicate. What is this metric? Accuracy? Loss? Perplexity?

MiniLM-based diversity is only one notion of coverage. It does not capture task-relevant diversity.

[1] Massive Activations in Large Language Models

**Questions:**

N/A

---

### Official Review · Reviewer_fuFt · 2025-10-28

**Soundness:** 3
**Presentation:** 3
**Contribution:** 2
**Rating:** 6
**Confidence:** 4

**Summary:**

The paper proposes to investigate neuron activations in pruning to understand whether pruning affects specific parts of NNs in specific tasks. Three SOTA pruning methods are used on several tasks and datasets with comprehensive experiments and analysis.

The idea of trying to understand pruning, the neuron activations and their relationship to semantics of models in different tasks is nice. The paper also resports rigorous experiments with many models and tasks. The downside is that the results stay at the level of "understanding" and there is limited evidence on using the results in practical pruning to guide the pruning process. There is a proposal for activation-guided pruning, but the empirical evidence to support and help understanding its performance remains superficial.

**Strengths:**

- Interesting empirical comparison of neuron activations

- Rigorous experiments

- Healthy set of models, calibration data, and datasets

- Somewhat interesting findings, in particular for various choices of pruning and their relation to semantic capability of the pruned model compared to an unpruned counterpar

**Weaknesses:**

- It is not surprising, and in fact is somehow known, that when neuron activations are not available (pruned or dropped), other parts of the network will replace them. This behavior has been previously found in LLMs.

- There is no clear metric from NSA that could directly guide pruning or help understand whether pruning has been done in a way that would hurt "semantic" -level performance of the pruned model

**Questions:**

- How can the results be used in practice to understand whether pruning has been successful or unsuccesful
- How can the approach be potentially used to guide pruning (new methods); this has been implied in several parts of the paper, but I had difficulties understanding how this could be directly applied in practical pruning settings.

Some parts of the paper are a bit unclear and remain at a superficial level, in particular for the performance of activation-guided pruning. I would like the authors to clarify this part to allow better evaluation of this aspect of their work.

---

### Official Review · Reviewer_gL2z · 2025-10-30

**Soundness:** 2
**Presentation:** 1
**Contribution:** 1
**Rating:** 0
**Confidence:** 5

**Summary:**

This paper evaluates three post-training LLM pruning methods (SparseGPT, Wanda, RIA) across 10 models, 24 datasets in four task types, revealing sharp performance drops tied to calibration data mismatches and semantic sensitivities. It introduces Neuron Semantic Attribution (NSA), a three-step interpretability framework - (1) clustering task-relevant tokens. (2) matching via activation scores. (3) comparing pre/post-pruning activations to quantify semantic losses. Contributions include pruning guidance, activation-guided regularization for low-activity sparsity, and bridging compression with explainability.

**Strengths:**

The work contributes practical advice on calibration and activation-guided pruning for better LLM compression, potentially aiding deployment in nuanced domains, but builds incrementally on recent studies without major paradigm shifts in explainable pruning.

**Weaknesses:**

1. The paper is written poorly, with typos (e.g., "perforamnce", "sumed activation"), inconsistent formatting (e.g., abrupt figure insertions), and vague details (e.g., undefined hyperparameters like $\lambda$ in SparseGPT).
2. Reproducibility Issues: No code release or detailed hyperparameters (e.g., SAE expansion factors, LLM prompts for token clustering) are mentioned, rendering NSA's implementation opaque; experiments use unspecified seeds, risking non-reproducible results in variable hardware setups.
3. Ethical Oversights: Ignores pruning's potential to amplify biases in sensitive tasks.

**Questions:**

1. Reproducibility Details for SAE Integration and Token Clustering: What are the exact prompts used for the LLM-guided token clustering in NSA Step 1, and how sensitive are results to prompt variations or different LLMs.
2. Statistical Rigor in Empirical Results: Table 1 reports averaged accuracies without error bars, confidence intervals, or significance tests.
3. Writing and Clarity Improvements: Typos (e.g., "perforamnce," "sumed activation") and vague phrasing (e.g., undefined λ in SparseGPT metric) detract from readability. A proofread revision with consistent notation and inline figure explanations would improve accessibility.
4. Prior Work on Interpretability for Pruning: The paper claims NSA is the "first" interpretability framework specifically for pruned LLMs linking activations to task-relevant semantics, but recent works like [1,2] already propose neuron-level interpretability via Integrated Gradients to identify and prune, i suggest including those works in related work.



[1] Detecting and Pruning Prominent but Detrimental Neurons in Large Language Models


[2] Mitigating Copy Bias in In-Context Learning through Neuron Pruning

---

### Official Review · Reviewer_yAN9 · 2025-11-01

**Soundness:** 2
**Presentation:** 2
**Contribution:** 2
**Rating:** 4
**Confidence:** 3

**Summary:**

This paper addresses the need for interpretability in LLM pruning by introducing neuron semantic attribution (NSA), a method that analyzes pruning effects through neuron-level semantics. Proposed NSA works by linking individual neurons to task-relevant concepts, thereby providing a fine-grained understanding of how pruning impacts model behavior and contributes to task-specific sensitivity. The paper conducts an empirical study involving 10 LLMs, 24 datasets, 4 major task categories, and various pruning configurations to examine pruning generalizability and robustness. The paper reveals that pruned models exhibit strong task-dependent sensitivity, particularly within sentiment classification and semantic similarity tasks, and further experiments verify that the choice of calibration data is a crucial factor in determining the generalizability of pruned models. The paper demonstrates that NSA serves as a reliable tool for interpreting and guiding pruning strategies, effectively helping to bridge the gap between model compression and interpretability.

**Strengths:**

- The paper presents a comprehensive empirical evaluation of LLM pruning, which significantly broadens the scope of existing literature for various setups.
- The extensive experiments successfully reveals strong task-dependent sensitivity in pruned models.

**Weaknesses:**

- While the systematic verification of the crucial role of calibration data in determining pruning generalizability is presented as a significant contribution, it can be argued that observing large performance variability based on the calibration set is somewhat expected, given that pruning methods rely on calibration data to select which parameters to remove. If pruning is inherently influenced by the sampled distribution, performance degradation on evaluation tasks outside that distribution is largely predictable.
- Although the paper explores the correlation between pruning effectiveness and the semantic coverage of the calibration data, a more explicit analysis detailing the performance differences based on the direct similarity or alignment between the calibration task and the evaluation task would strengthen the paper.
- The paper emphasizes the proposed method, NSA overall paper. However, the analysis leveraging NSA appears limited. Beyond showing fragmented instances of activation decreases for key tokens, the paper lacks a more in-depth, systematic analysis that uses NSA across the entire range of tasks and calibration configurations studied to explain quantitatively how specific semantic losses relate to the accuracy drops observed in the large empirical tables. I suggest integration of NSA results with the comprehensive performance metrics would significantly enhance the interpretability framework's contribution and provide stronger evidence for its reliability in guiding future pruning strategies.

**Questions:**

- As stated at the weaknesses, I expect the more deeper analysis using the proposed NSA and the deeper analysis on the relation between the calibration task and the evaluation task.

---

### Meta-Review · Area_Chair_wWGo · 2026-01-07

**Summary:**

The paper presents a large-scale empirical study of post-training pruning in large language models and introduces Neuron Semantic Attribution (NSA), an interpretability framework that links neuron-level activation changes caused by pruning to task-relevant semantics. The main strengths are the broad experimental coverage and the consistent empirical observation (that pruning sensitivity strongly depends on task type and calibration data). However, multiple reviewers challenged that the interpretability contribution does not convincingly go beyond existing work, and the current evidence is insufficient to support the paper’s stronger claims.

**Reviewer Concerns:**

The authors didn't post any response to the reviews.

**Reviewer Scores:**

The authors didn't post any response to the reviews.

---

### Decision · Program_Chairs · 2026-01-26

Reject